# Synthesis and Characterization of Healable Waterborne Polyurethanes with Cystamine Chain Extenders

**DOI:** 10.3390/molecules24081492

**Published:** 2019-04-16

**Authors:** Dae-Il Lee, Seung-Hyun Kim, Dai-Soo Lee

**Affiliations:** Division of Semiconductor and Chemical Engineering, Chonbuk National University, Baekjedaero 567, Deokjin-gu, Jeonju, Chonbuk 54896, Korea; ldi4084@naver.com (D.-I.L.); thdnlsla@naver.com (S.-H.K.)

**Keywords:** waterborne polyurethane, self-healing, cystamine, disulfide, metathesis

## Abstract

In this study, environmentally friendly, self-healing waterborne polyurethanes (WPUs) were prepared based on the disulfide metathesis reaction in cystamine. The cystamine acted as a chain extender in the WPU film, which showed a high mechanical strength of 19.1 MPa. The possibility of self-healing reaction was simultaneously modeled via liquid chromatography–mass spectrometry (LC-MS). WPU was confirmed to self-heal a surface crack thermally after a scratch test, and the efficiency was measured by comparing the mechanical properties before and after a cut-and-healing test. In addition, the disulfide-thiol exchange reaction was confirmed to occur in WPU with cystamine as a chain extender and 2-mercaptoethanol. Hot press tests confirmed the possibility of reprocessing the WPU. The WPU incorporating disulfide groups showed great potential as a smart self-healing material.

## 1. Introduction

Waterborne polyurethanes (WPUs) are attracting attention as environmentally friendly synthetic materials. The method used for dissolving polyurethanes in existing organic solvents poses environmental problems. WPUs were developed to solve the problems caused by volatile organic compounds (VOCs); ionic groups are introduced into the polyurethane chain and these compounds are dispersed in water [1,2]. Such environmentally friendly, multi-functional WPUs are commonly used in a wide range of applications, including coatings [3] and adhesives [1,4]. The external exposure environments of these materials cause contact cracks, friction, continuous impact, and abrasion, resulting in fine cracks on the polymer surface. These cracks considerably impact the durability and lifespan of the polymer. Self-healing of such fine cracks could increase the lifespan and durability of polymers. Therefore, many researchers are striving to introduce self-healing functions into WPUs as a way to increase their lifetime [5,6,7,8,9,10,11,12]. Tensile strength and healing-efficiency of some previously reported WPUs are given in Appendix A.

Self-healing polymers have recently received attention as breakthrough materials that can repair their own cracks caused by external impacts; these polymers can maintain a long lifetime and durability and can even be recycled [13]. The self-healing properties of polymers can be implemented by intrinsic or extrinsic properties [14]. Self-healing by intrinsic properties is imparted through the dynamic nature of physical interactions or the introduction of reversible covalent bonds in the polymer chain. Conversely, external self-healing is conferred by healing agents that are intentionally pre-embedded in microcontainers [15,16]. Intrinsically self-healing polymers can be fabricated via Diels–Alder reactions [5] as well as by introducing disulfide bonds [17,18,19], hydroxyl groups [20], imine groups [21], boric acid [22], hindered ureas [23], hydrogen-bonding interactions [24], metal ligands [25], and various additives [6,7] into polymers to impart self-healing behavior. In these methods, reversible interactions can be triggered by several external factors (such as heat, moisture, UV, acids, and bases). Among these approaches, using a disulfide group is of particular research interest. Disulfide bonds have lower binding energies than C–C and C–H bonds in many molecules and are classified as weak bonds. These weak bonds can be broken via heating, mechanical stress, and light irradiation, leading to the aggressive formation of thiolate and new disulfide bonds [26]. Thiol–disulfide exchange has been previously reported to occur via a radical mechanism [27]. Therefore, introducing disulfide into polymers can induce self-healing properties [28,29,30]. Nevejans et al. introduced an aromatic disulfide group into a WPU film; after drying, the film exhibited high mechanical strength and self-healing properties [31,32]. Introduction of aromatic disulfide groups into WPUs improves the self-healing performance of WPUs, however, when WPUs are used as a coating material, the material surface turns yellow under environmental exposure due to the presence of aromatic groups, therefore some means to reduce this yellowing of the WPU must be realized.

This paper presents a simple pathway for designing and manufacturing self-healing WPUs based on aliphatic disulfide reactions. The prepolymer was prepared using the conventional method for manufacturing WPUs, i.e., by reacting an isocyanate with both terminal ends of a polyol and then neutralization and dispersion of the product in water. Cystamine was used as chain extender to prepare disulfide group-bearing WPUs and their self-healing behaviors were compared. We also confirmed that the disulfide reaction occurs via a reaction with monothiol, thus revealing the recyclability of this material. Finally, we confirmed the self-healing performance via a surface scratch healing test and cut-and-healing test.

## 2. Results and Discussion

Figure 1 schematizes the morphology of microphase separated films for EDA-chain-extended WPU (WPU-EDA) and cystamine-chain-extended WPU (WPU-cystamine), which reveals the role and location of the introduced disulfide groups. Conventionally, polyurethane shows a microphase separation into hard and soft segments. The hard segments are comprised of segments of the polyurethane structure that are capable of hydrogen bonding, which may include urethane, urea groups, or benzene rings via van der Waals interactions. 

In contrast, the soft segment is composed of flexible parts of the polyurethane structure with weak bonding strength, mainly the polyol part. The disulfide group in this study is introduced through cystamine, and it functions as a chain extender of the prepolymer. In addition, it forms a urea group on both sides and is located in the hard segment. Thus, even if a disulfide group with a weak bonding force is introduced, good mechanical properties can be attained due to the hydrogen-bonding force within the hard segments [33,34]. However, for self-healing to occur, the temperature at which the hydrogen-bonding force is weakened must exceed the glass transition temperature of hard segments, which is a disadvantage.

Figure 2a shows the disulfide metathesis reaction. Heat-disrupted disulfides form thiolates, which attack other disulfides to form new disulfide groups. Experiments with model compounds, Disulfide 1 and Disulfide 2, were performed using liquid chromatography–mass spectrometry (LC-MS) to confirm this disulfide metathesis. As shown in Figure 2b, two kinds of disulfide compounds with different molecular weights were dissolved in water and the reaction was carried out at 80 °C for 24 h to ensure that the disulfide metathesis reaction proceeded sufficiently. Figure 2c shows the formation of the product (M_n_ 197 g mol^−1^) of metathesis reaction between Disulfide 1 (M_n_ 154 g mol^−1^) and Disulfide 2 (M_n_ 240 g mol^−1^). The LC-MS spectra clearly confirm the disulfide metathesis reaction. All the products formed via disulfide metathesis reaction were also confirmed by their ^1^H-NMR spectra (Appendix A).

Figure 3 shows the tensile stress–strain curves of WPU chain extended with EDA or cystamine. WPU-EDA has high mechanical strength of 26.5 MPa and a high maximum strain value of 516%. In contrast, the mechanical strength of WPU-cystamine is 19.1 MPa, and maximum strain is 323%. As the molecular weight of WPU-cystamine is not so much different from that of WPU-EDA (Appendix A), the different physical properties are attributable to the characteristics of cystamine. 

These results indicate that the disulfide groups, which are weakly bonded to each other, cannot sustain a mechanical force and break at the center of the hard segment, leading to weak mechanical properties. However, unlike similar materials in other papers [28], WPU-cystamine shows adequate mechanical properties for practical applications.

Figure 4 shows the DSC thermograms of WPU films. Glass transition temperatures of soft segments (T_gs_) and hard segment (T_gh_) for WPU-EDA and WPU-cystamine were observed around −80 °C and above 50 °C, respectively. Appendix A show the dynamic mechanical properties of WPU films. Glass transition temperatures (T_g_s) of polymers were determined from the peak temperatures of loss moduli in DMA (Appendix A). Table 1 lists the T_g_s for WPU films determined via DSC and DMA.

The self-healing process of WPU film was investigated through scratch tests, which were separately conducted at 110 °C and 130 °C, as shown in Figure 5 and Figure 6, respectively. Specifically, the upper surface of the WPU film was scratched with a razor blade, followed by heat treatment for 0, 1, and 3 h at each temperature, which initiated a disulfide metathesis reaction. Scratches on the WPU-EDA film apparently remain intact during the heat treatments at 110 °C and 130 °C. However, in the case of WPU-cystamine, the crack was observed to close slightly during the heating process at 110 °C. Thus, scratches partially disappear, confirming the possibility of self-healing via the disulfide metathesis reaction. Moreover, scratches significantly disappear after only 1 h of heat treatment at 130 °C. This finding reveals that the surface of the damaged film can be restored via the disulfide metathesis reaction.

As shown in Figure 7 and Table 2 and Appendix A, the efficiency of the self-healing process of WPU films after cut-and-healing tests was systematically investigated using Equation (1):Self-healing efficiency = σ_healed_/σ_uncut_ × 100%(1)
where σ_healed_ and σ_uncut_ are the tensile strengths of self-healed and uncut (as synthesized) samples, respectively. The self-healing behavior was confirmed by measuring the mechanical properties of the WPU films before and after cut-and-healing tests. The films were formed into a dog bone shape and then completely cut into two pieces using a razor blade. The two separate slices were then brought into contact in two convection ovens at 110 °C and 130 °C to observe the change in the self-healing efficiency with time. A maximum efficiency of 28% was achieved after 3 h of heat treatment at 110 °C; the efficiency decreased thereafter. A high efficiency of up to 40% was observed after 3 h of heat treatment at 130 °C. These findings confirm that the self-healing efficiency of the WPU system based on the disulfide bonds of cystamine is maximized after 3 h at 130 °C. Beyond 3 h at elevated temperatures in the convection oven, the self-healing efficiency might decline due to the weakening of the hard segment packing via hydrogen-bond breakage caused by exchange reactions in the specimens. As shown in Appendix A, the heat treatment lowered the T_gh_s of the WPU films. The effects of the thermal treatments are summarized in Appendix A. Weakening of the hydrogen bonds in the hard segments appeared in the FT-IR spectra of the heat-treated WPU films (Appendix A). Before the heat treatment, hydrogen-bonded carbonyl groups of urethane and urea are observed at 1695 and 1637 cm^−1^, respectively, in the FT-IR spectra of the WPU films. After heat-treating the WPU films at 110 °C and 130 °C, the carbonyl peak of urea was decreased whereas the free-carbonyl peak increased, thereby shifting the urethane carbonyl peak [35,36]. The heat treatment of WPU films also accompanied slight yellowing with time as shown in Appendix A. The yellowing of the WPU films is attributable to the side reactions during the heat treatments in the convection oven. It is speculated that the side reactions also decline the self-healing efficiency of WPU films.

The exchange reaction of the disulfide group of WPU-cystamine by a thiol could be directly confirmed. One way to confirm this reaction is to use 2-mercaptoethanol. As shown in Figure 8a, a piece of each type of WPU film was placed in a bottle with 2-mercaptoethanol and the disintegration via the exchange reaction with 2-mercaptoethanol was then activated at 80 °C for 2 h. 

The shape of the WPU-EDA film was retained as shown in the left bottle of Figure 8a. In contrast, the WPU-cystamine film evidently degraded due to the exchange reaction of disulfide groups because the thiol group of the 2-mercaptoethanol reduces the disulfide group of WPU-cystamine, which turned into small molecules, as shown schematically in Figure 8b. The results show that the disulfide group of WPU-cystamine can also undergo an exchange reaction, thus demonstrating the possibility of being recycled using small thiol compounds.

Figure 9 shows the recyclability of the WPU-cystamine films estimated by repeating the heat and pressure applications. When the WPU-cystamine films are pulverized and then subjected to high temperature and pressure, a disulfide metathesis reaction occurs between the surfaces. In this way, several pieces can be reprocessed into specimens for the tensile tests. After all repeated recycling tests, the tensile strength of WPU-cystamine recovered to nearly 85% (Figure 9b). However, WPU-EDA could not be reprocessed under the same conditions (Appendix A).

## 3. Experimental

### 3.1. Materials

Poly(tetramethylene ether glycol) (PTMEG, molecular weight (M_n_) = 2000 g mol^−1^) was purchased from Sigma-Aldrich (Yong-In, Korea), and it was placed in a vacuum oven at 60 °C for 1 day to remove moisture before use. 2,2-Bis(hydroxymethyl) propionic acid (DMPA) and 2-hydroxyethyl disulfide were also purchased from Sigma-Aldrich. Isophorone diisocyanate (IPDI), triethylamine (TEA), ethylenediamine (EDA), 2-mercaptoethanol, and L-cystine were purchased from Daejung Chemical (Si-Heung, Korea), whereas cystamine dihydrochloride was purchased from Tokyo Chemical Industry (Tokyo, Japan). All compounds were used without further purification.

### 3.2. Synthesis of Waterborne Polyurethane

WPU films were prepared by prepolymer production, dispersion, and film drying in three stages. Acetone was used to dissolve DMPA and decrease the viscosity of the prepolymer. Scheme 1 shows the detailed synthesis process while Table 3 shows the recipes for synthesizing WPU-EDA and WPU-cystamine. 

To prepare the prepolymer, PTMEG, DMPA, and IPDI were added to the reactor and stirred at 60 °C for 3 h under a nitrogen atmosphere. The isocyanate groups of IPDI were allowed to react until the theoretical value of the isocyanate content for the prepolymer was reached. The isocyanate content was determined during the reaction using the ASTM D1638-74 method. At this point, the temperature of the reactor was reduced to room temperature and enough moles of TEA were added to neutralize the carboxyl groups of DMPA by stirring for 30 min. To disperse the neutralized prepolymer, the mixture was then stirred at 600 rpm and deionized water (solid contents 20%) was added. EDA or cystamine, which were separately used as chain extenders, were dissolved in a small amount of water and added dropwise into the dispersion. After all of the EDA or cystamine was added, the temperature of the reactor was raised to 60 °C, and reaction proceeded with stirring for 3 h. After stirring for 3 h, WPU dispersion was attained. The particle size distributions of WPUs are shown in Appendix A, and the average particle sizes are listed in Appendix A. Even though WPU-cystamine showed the presence of large particles, the dispersion did not lose the stability for three months. WPU films were prepared by casting the respective dispersions in a Teflon mold and drying them in a 30 °C convection oven for one week to prevent fine bubble formation and in a vacuum oven for one day.

### 3.3. Confirmation of the Disulfide Metathesis Reaction

To confirm that the disulfide metathesis reaction proceeded successfully, a piece of each type of WPU film was separately placed in a bottle and enough mercaptoethanol was added to fully submerge the film. Thereafter, the disulfide metathesis reaction was activated by heating the bottles to 80 °C in a convection oven for 2 h.

### 3.4. Characterization

LC/MS (AGIL ENT 1100, Agilent Technologies, Palo Alto, CA, USA) was used to model the disulfide metathesis reaction in WPU. The mechanical strength and self-healing efficiency of films after cut-and-healing tests were measured on a universal testing machine (UTM, LR5K plus, Lloyd Instruments, West Sussex, UK). During the test, the center of the dog bone specimens were cut using a razor blade and the cut surfaces were contacted each other to promote self-healing at the 110 °C and 130 °C in the oven. The tensile properties of the samples were measured at 25 °C by pulling at 500 mm/min. All samples were measured with three specimens, and the average values were obtained. A thin scratch was applied to the fixed sample surface using a razor blade. The sample surface cut with razor blade had thin visible scratches. An image of the scratch on the surface was obtained using a scanning electron microscope (SEM, AIS2100, Seron Technologies Inc., Seongnam, Korea). Then, the sample surface was coated using an ion coater (HC-21, Hoyeontech, Seongnam, Korea). Differential scanning calorimetry (DSC, Q 20, TA Instruments, New Castle, DE, USA) was used to measure the films’ thermal properties from −100 to 150 °C at a heating rate of 10 °C/min in the presence of nitrogen gas. The dynamic mechanical properties were investigated employing a dynamic mechanical analyzer (DMA, Q 800, TA Instruments) from −100 to 200 °C at a heating rate of 5 °C/min. The structure of the model compound was analyzed using ^1^H-NMR spectra obtained on a 600 MHz FT-NMR spectrometer (JNM-ECA600, JEOL Ltd., Tokyo, Japan). All samples were dissolved in deuterium oxide (D_2_O) and spectra measured. Fourier transform-infrared (FT-IR) spectra of the samples before and after the heat treatment were also studied in attenuated total reflection mode using FT/IR 300E (JASCO, Tokyo, Japan).

## 4. Conclusions

By introducing cystamine into the polymer backbone of a WPU, we have designed and synthesized a new kind of WPU that can exploit the disulfide metathesis reaction. The WPU-cystamine film showed excellent self-healing behavior. A scratch test showed cracks disappearing at 130 °C within 1 h. The self-healing behavior was investigated through a cut-and-healing test, the efficiency found to be 40% at 130 °C for 3 h. However, the disulfide groups were located inside the hard segment due to the addition as a chain extender; thus, the self-healing efficiency was not high because molecular chains were not free to diffuse and migrate. However, unlike other studies in which aliphatic disulfide groups were added to WPU, the WPU-cystamine sample in this study showed a high mechanical strength of 19.1 MPa. This research thus enables the possibility of increasing the durability and lifetime of environmentally friendly WPU via self-healing abilities.

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
