# Peer review of "Synthesis and Characterization of Healable Waterborne Polyurethanes with Cystamine Chain Extenders"

_molecules, 2019, doi:10.3390/molecules24081492_

Round 1

Reviewer 1 Report

This paper deal with the use of cystamine as a disulfide containing chain extender to generate self-healing polyurethanes from waterborne dispersions. Although the manuscript is of interest a number of things should be clarified before publication:

A few sections of the paper miss several key citations:

1) The introdcution claims only weak self-healing polymers have been made from polyurethane dispersions. Hoewver, high strength polyurethane dispersions with self healing properties have recently been reported by Nevejans et al. (see European Polymer Journal 2019, 112, 411-422 and Polymer 2019, 166, 229-238) and these should be referenced.

2) The mechanism for disulfide exchange in the paper is suggested to be by thiol/disulfide exchange at high temperatures. However, previous work suggests that a radical mechanism may also play an important role (Physical Chemistry Chemical Physics 18 (39), 27577-27583).

3) A number of studies have shown substantially improved self healing through incorporation of the disulfide as is also reported here into the hard phase of the material and should also be referenced. (Macromol. Chem. Phys. 2017218, 1600320, Adv. Mater. 201830, 1802556)

Some points in the discussion need further expansion:

5) The material stops increasing in self healing behavior above 130 degrees. The authors attribute this to phase rearrangement of the hard phase but do they discount the possibility that at this temperature side reactions occur that limit recovery of mechanical strength? Do they see color change for example

6) In Figure 9a the reprocessed sample still retains some of the original heterogeneous shape. This suggests increased time/temperature is required to properly form a sample that would be suitable for tensile testing (i.e. with uniform thickness) Can the authors confirm that the tensile sample was homogeneous?

7) The particle size of the disulfide sample was signifcantly higher than the other sample and also shows the presence of large particles (micron sized). The authors should comment on the stability of the dispersions as this suggests the cystamine samples were poorly stabilized.

Author Response

Response to Reviewer 1 Comments

1) The introduction claims only weak self-healing polymers have been made from polyurethane dispersions. However, high strength polyurethane dispersions with self-healing properties have recently been reported by Nevejans et al. (see European Polymer Journal 2019, 112, 411-422 and Polymer 2019, 166, 229-238) and these should be referenced.

Previous studies by Nevejans et al were described and the necessity of our work was mentioned in the Introduction of the revised manuscript (Line 53 -60).

2) The mechanism for disulfide exchange in the paper is suggested to be by thiol/disulfide exchange at high temperatures. However, previous work suggests that a radical mechanism may also play an important role (Physical Chemistry Chemical Physics 18 (39), 27577-27583).

In the revision, additional possible mechanism for disulfide metathesis was introduced (Line 51-52).

3) A number of studies have shown substantially improved self-healing through incorporation of the disulfide as is also reported here into the hard phase of the material and should also be referenced. (Macromol. Chem. Phys. 2017, 218, 1600320, Adv. Mater. 2018, 30, 1802556)

Previous studies in the literatures on self-healing through incorporation of the disulfide into the hard phase were mentioned in the revision (Line 141-143).

5) The material stops increasing in self-healing behavior above 130 degrees. The authors attribute this to phase rearrangement of the hard phase but do they discount the possibility that at this temperature side reactions occur that limit recovery of mechanical strength? Do they see color change for example?

In the revision, photographs of specimens after heat treatments were given in supporting information and the possibility of side reactions that limit recovery of mechanical strength was discussed (Figure S12, Line 218-222).

6) In Figure 9a the reprocessed sample still retains some of the original heterogeneous shape. This suggests increased time/temperature is required to properly form a sample that would be suitable for tensile testing (i.e. with uniform thickness) Can the authors confirm that the tensile sample was homogeneous?

The specimens for tensile tests were homogeneous as shown in Figure 9 in the revision. We took the pictures of the reprocessed samples again to confirm that the samples were homogenous.

7) The particle size of the disulfide sample was significantly higher than the other sample and also shows the presence of large particles (micron sized). The authors should comment on the stability of the dispersions as this suggests the cystamine samples were poorly stabilized.

Even though WPU-cystamine showed the presence of large particles, the dispersion did not lose the stability for three months. The stability of the dispersions was mentioned in the revised manuscript (Line 94-95).

Reviewer 2 Report

The authors mainly compared two WPU polymeric materials for screening self-healing properties. The one with cystamine chain extender exhibits better self-healing properties although the mechanical strength is lower than the counterpart. However, there are several parts the author need to clarify before the acceptance.

1.       Very first. Why do the authors do this piece of research? I cannot find the clue from introduction? I believe the authors may need rewrite the introduction.

2.       The polymer weight-dependent mechanical properties, glass transition temperature and self-healing performance should be demonstrated.

3.       Comparison between other reported WPU materials should be done.

Author Response

Response to Reviewer 2 Comments

1) Very first. Why do the authors do this piece of research? I cannot find the clue from introduction? I believe the authors may need rewrite the introduction.

In the revision, previous studies on self-healing WPU were described additionally and the necessity of our work was mentioned in the Introduction (Line 34-35, 51 -60).

2) The polymer weight-dependent mechanical properties, glass transition temperature and self-healing performance should be demonstrated.

The molecular weights of the WPU polymers were given in Table S3 and discussed in the revised manuscript (Line 162-164).

3) Comparison between other reported WPU materials should be done

In the revised manuscript, tensile strength and self-healing efficiency of self-healing WPUs previously reported in the literature were summarized in Table S1 and mentioned in the Introduction of manuscript (Line 34-35)

Round 2

Reviewer 2 Report

The manuscript can be accepted now, although the introduction part is still not quite clear.